# Increasing metabolic co-morbidities are associated with higher risk of advanced fibrosis in nonalcoholic steatohepatitis

**Robert J. Wong**[1]*, **Tram Tran**[2], **Harvey Kaufman**[3], **Justin Niles**[3], **Robert Gish**[4]

**1** Division of Gastroenterology and Hepatology, Alameda Health System–Highland Hospital, Oakland, CA, United States of America, **2** Gilead Sciences, Foster City, CA, United States of America, **3** Quest Diagnostics, Secaucus, NJ, United States of America, **4** Division of Gastroenterology and Hepatology, Stanford Health Care, Palo Alto, CA, United States of America

* Rowong@alamedahealthsystem.org

**Data Availability Statement:** The data underlying the results presented in the study are available from Quest Diagnostics Clinical Laboratory Database and are stored in the Quest Diagnostics

## Abstract

Hepatic fibrosis and advanced fibrosis in particular is the strongest predictor of liver-related outcomes and mortality among nonalcoholic steatohepatitis (NASH) patients. Understanding prevalence and predictors of NASH with advanced fibrosis is critical for healthcare resource planning. Using a large U.S. clinical laboratory database from 10/1/2017-9/30/2018, adults negative for hepatitis B and hepatitis C and after excluding for alcoholic liver disease and pregnancy were evaluated for prevalence of F3 and F4 fibrosis using a systematic algorithm of five fibrosis-4 (FIB-4) criteria: Criteria 1 ($\geq$F3: >2.67), Criteria 2 (2.67<F3$\leq$4.12 and F4>4.12), Criteria 3 (2.67<F3$\leq$3.15, F4>3.15), Criteria 4 (3.25<F3$\leq$3.5, F4>3.5), Criteria 5 (3.25<F3$\leq$4.12, F4>4.12). Metabolic co-morbidities evaluated included decreased high density lipoprotein (<40 mg/dL men, <50 mg/dL women), high triglycerides ($\geq$150 mg/dL), elevated hemoglobin A1C ($\geq$6.5%). Parallel analyses of patients with specific NAFLD/NASH ICD-9/10 codes from 10/1/2013-9/30/2018 were performed. Multivariate logistic regression models evaluated for predictors of $\geq$F3 fibrosis. Among patients with NAFLD/NASH ICD-9/10 codes, $\geq$F3 prevalence ranged from 4.35% - 6.90%, and F4 prevalence ranged from 2.52%– 3.67%. Increasing metabolic co-morbidities was associated with higher risk of $\geq$F3 fibrosis. Compared to NASH patients without metabolic co-morbidities, NASH with four concurrent metabolic co-morbidities had higher risk of $\geq$F3 (OR 1.56, 95% CI 1.40–1.73, p<0.001). In summary, prevalence of NASH with advanced fibrosis among U.S. adults was as high as 6.90% and prevalence of NASH with cirrhosis was as high as 3.67%, representing 5.18 million and 2.75 million, respectively, when using an estimate of 75 million U.S. adults with NAFLD. Co-morbid metabolic abnormalities were associated with higher risk of advanced fibrosis among NASH patients.

Informatics Data Warehouse (https://www.
questdiagnostics.com/home/physicians/
healthcareit/quanumsolutions/informaticsdatafeed.
html).

**Funding:** This study was supported by a research
grant from Gilead Sciences. The funders had no
role in study design, data collection and analysis,
decision to publish, or preparation of the
manuscript.

**Competing interests:** I have read the journal's
policy and the authors of this manuscript have the
following competing interests: Robert Wong:
research grants, consulting, advisory board,
speaker's bureau – Gilead Sciences; research
grants – Abbvie; speaker's bureau – Salix, Bayer;
research grant – AASLD Foundation; Tram Tran:
employee and stocks – Gilead Sciences; Harvey
Kaufman: None; Justin Niles: None; Robert Gish:
consulting, advisor - Abbot, AbbVie, Alexion,
Arrowhead, Bayer AG, Biocollections, Bristol-Myers
Squibb Company, Contravir, Eiger, Enyo,
eStudySite, Genentech, Gilead Sciences, HepaTX,
HepQuant, Hoffmann-LaRoche Ltd., Intellia,
Intercept, Ionis Pharmaceuticals, Janssen,
MedImmune, Merck, Prometheus, Quest,
Shionogi, Transgene, Trimaran; Scientific/clinical
Advisory Boards -AbbVie, Merck, Arrowhead,
Bayer, Contravir, Dova Pharmaceuticals, Eiger,
Enyo, Janssen, Medimmune, Janssen/J&J,
Intercept, Shionogi, Spring Bank; Clinical trials –
eStudySite. Chair Clinical Advisory Board –
Arrowhead; Data Safety Monitoring Board – Ionis;
Speaker's bureau – AbbVie, Alexion, Bayer, BMS,
Gilead Sciences Inc., Merck; Minor stock
shareholder - Athenex, Triact, Synageva,
RiboSciences, CoCrystal; Stock options –
Arrowhead, Eiger. The above competing interests
statement and funding do not alter our adherence
to all PLOS ONE policies on sharing data and
materials.

## Introduction

Nonalcoholic steatohepatitis (NASH) is a leading cause of chronic liver disease in the United States and globally[1, 2], and is rapidly becoming a leading cause of end-stage liver disease leading to hepatocellular carcinoma (HCC) and need for liver transplantation[3–5]. The presence of fibrosis is the strongest predictor of mortality and liver-related complications among patients with NASH[6–8]. Advanced fibrosis in particular, which is the subset of patients with F3 or greater fibrosis, have significantly greater risks of liver-related complications, risk of HCC, and risk of liver-related and all-cause mortality in NASH patients[6, 8]. Earlier identification of NASH patients with advanced fibrosis has clinical significance, given that these patients need closer monitoring including screening for HCC or varices, more aggressive management of risk factors, and are at greatest need for NASH therapies to halt or reverse steatohepatitis and fibrosis.

The prevalence of NASH with advanced fibrosis is not clear and current estimates are likely underestimates given that existing studies have utilized primarily observation cohort-type or survey-based study designs that underestimate the true prevalence[1, 9, 10]. In addition, modeling studies or systematic reviews aimed at defining NASH and NASH with advanced fibrosis prevalence are based on existing literature, which may be biased given that NASH is likely underdiagnosed given sub-optimal provider and patients awareness, the lack of clear guidelines on who to screen and what tools to screen with, the lack of effective therapies that often drives disease screening efforts, as well as the lack of large datasets with adequate data to assess hepatic fibrosis[11–13]. More accurately understanding prevalence and predictors of advanced fibrosis among NASH patients is important to guide healthcare resource planning such that adequate preventative care and interventions can be effectively targeted to those NASH patients at greatest risk of liver-related complications. To address this gap in knowledge, we aim to utilize a nationally representative U.S. clinical laboratory database to evaluate prevalence of advanced fibrosis using a systematic algorithm of non-invasive fibrosis biomarker (fibrosis-4 (FIB-4)) and to evaluate the impact of metabolic co-morbidities on odds of advanced fibrosis.

## Materials and methods

A retrospective cohort study was performed using clinical laboratory data from October 1, 2017 to September 30, 2018 from the Quest Diagnostics Clinical Laboratory Database. Quest Diagnostics has over 145 million patient encounters each year across the United States. Test results are stored in the Quest Diagnostics Informatics Data Warehouse, which is the largest private clinical laboratory data warehouse in the United States and stores approximately 3 billion test results annually. For this Quest Diagnostics Health Trends study, we extracted testing data for individual patients as described below; all data were de-identified prior to analysis. This study was determined to be exempt from institutional review by the Alameda Health System Institutional Review Board.

The study cohort was developed by first identifying unique adults (age $\geq$ 18 years) with unique identification numbers in the dataset. We then excluded patients with positive hepatitis B virus surface antigen (HBsAg) or positive hepatitis C virus antibody (HCV Ab), as well as patients with unavailable labs to assess FIB-4 (Fig 1). We further excluded patients with alcohol use or alcoholic liver disease related ICD-9/10 codes from 10/1/2013–9/30/2018 and patients with maternal screening at a time that indicated pregnancy when laboratory values were obtained. Among this cohort, prevalence of F3 and F4 fibrosis was assessed using a systematic algorithm consisting of five FIB-4 score criteria based on published literature[14–16]: Criteria 1 ($\geq$F3: >2.67), Criteria 2 (2.67<F3$\leq$4.12 and F4>4.12), Criteria 3 (2.67<F3$\leq$3.15 and

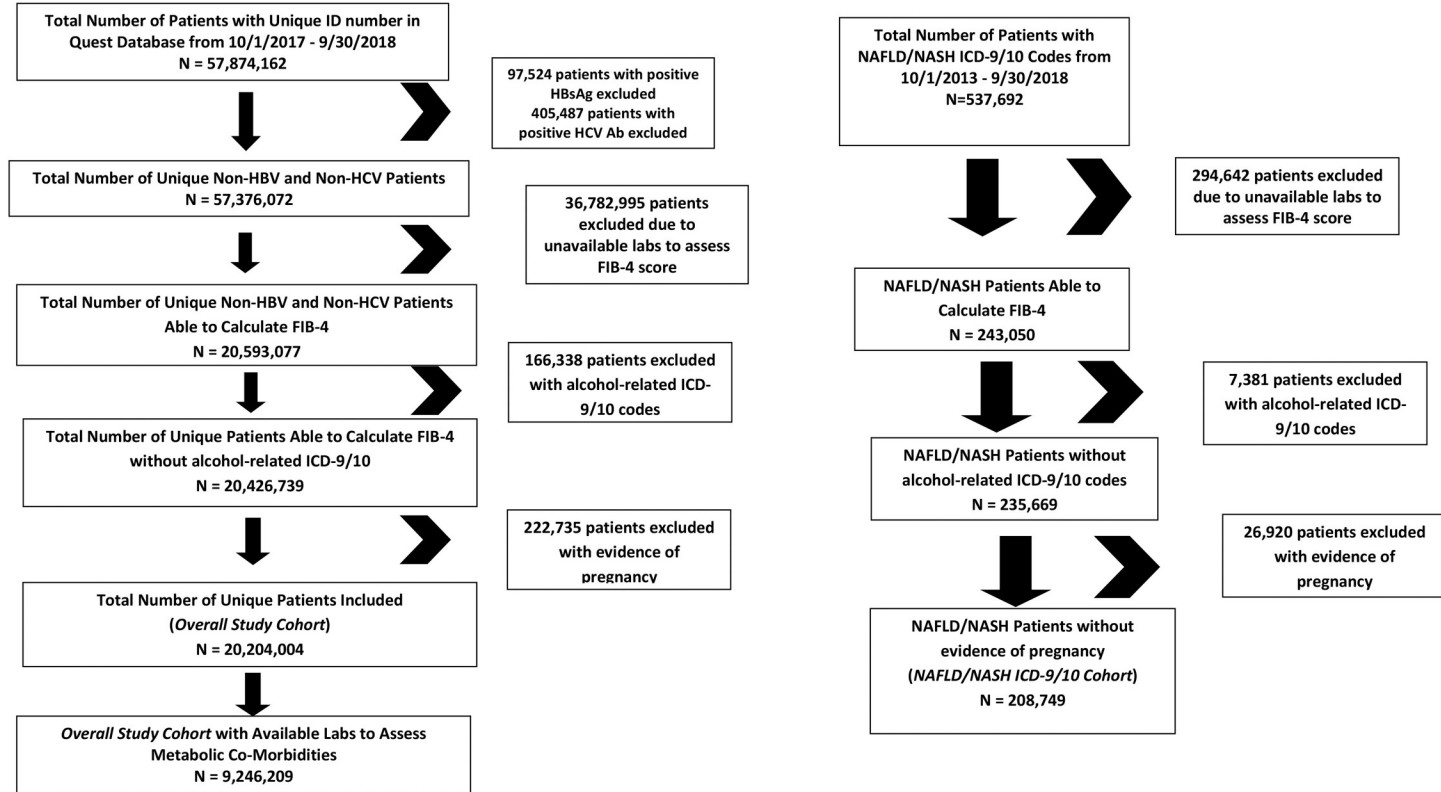

**Fig 1. Flowchart to identify the study cohort.**

F4>3.15), Criteria 4 (3.25<F3≤3.5 and F4>3.5), Criteria 5 (3.25<F3≤4.12 and F4>4.12). Given our focus on the impact of metabolic co-morbidities on the prevalence of advanced fibrosis, we focused on three laboratory biomarkers(decreased high density lipoprotein (HDL <40 mg/dL in men, <50 mg/dL in women), elevated triglycerides (TG ≥150 mg/dL), and elevated hemoglobin A1C (≥6.5%)) and at least one ICD-9/10 code for hypertension from 10/1/2013–9/30/2018 that are included components of the metabolic syndrome[17, 18]. Prevalence of F3 and F4 fibrosis were calculated in patients with elevated alanine aminotransferase (ALT > 25 U/L women, ALT > 35 U/L men) and one, two, three, or four concurrent metabolic co-morbidities. In addition to the above study cohort, we performed a parallel analysis specifically focusing on patients with NAFLD/NASH ICD-9/10 codes. To generate a larger sample for this parallel analysis, we included patients with NAFLD/NASH ICD-9/10 codes from 10/1/2013–9/30/2018 (Fig 1). Patients with unavailable labs to assess FIB-4, alcohol use or alcoholic liver disease-related ICD-9/10 codes, and patients with evidence of pregnancy were similarly excluded. Histologic data from liver biopsy and radiographic data from imaging-based studies were not available for inclusion in the study. Thus our use of the term NASH to refer to patients with F3 or F4 fibrosis is based on the assumption that those patients with suspected NAFLD and advanced fibrosis must have progressed to the NASH disease state[13].

Prevalence of F3 and F4 fibrosis using each of the above Criteria were calculated and presented as prevalence (%) with 95% confidence intervals (95% CI). Adjusted multivariate logistic regression models were utilized to evaluate predictors of advanced fibrosis (F3/F4 using Criteria 1). The final multivariate model adjusted for sex, age, and number of metabolic co-morbidities. In the final model, age is a continuous variable with a unit constituting 10 years.

Statistical analyses were performed using SAS Studio 3.6 on SAS 9.4 (SAS Institute Inc., Cary, NC, USA). A two-tailed p<0.05 indicated statistical significance. This study was granted exempt status from Alameda Health System Institutional Review Board.

## Results

From October 1, 2017 to September 30, 2018, 57,874,162 patients with unique identification numbers were identified in the database. After excluding patients with evidence of positive HBsAg or positive HCV Ab, we additionally excluded patients with unavailable labs to calculate FIB-4 scores, those with alcohol use or alcoholic liver disease related ICD-9/10 and patients with evidence of pregnancy (Fig 1). The final overall study cohort consisted of 20,204,004 patients. Among this cohort, 9,246,209 patients had available labs to assess metabolic co-morbidities. We also performed a parallel analysis that specifically focused on patients with NAFLD/NASH ICD-9/10 codes from October 1, 2013 to September 30, 2018, and after excluding patients as previously described, the final NAFLD/NASH ICD-9/10 cohort consisted of 208,749 patients (Fig 1).

Among the overall study cohort, 57.4% were female and mean age was 54.6 years (SD 17.3), 7.4% had elevated ALT, 29.0% had elevated triglycerides, 19.9% had elevated hemoglobin A1C, 24.3% had deceased LDL, and 39.1% had hypertension (Table 1). Overall prevalence of F3/F4 fibrosis using Criteria 1 was 3.12% (95% 3.11–3.13) (Table 2). Increasing number of metabolic co-morbidities was associated with higher prevalence of F3/F4 fibrosis using Criteria 1. For example, prevalence of F3/F4 fibrosis was 5.07% (95% CI 4.98–5.16) in patients with elevated ALT and one metabolic co-morbidity, which increased to 6.68% (6.44–6.92) among patients with elevated ALT and four metabolic co-morbidities, p<0.01. Among the NAFLD/NASH ICD-9/10 cohort, prevalence of F3/F4 fibrosis was 6.90% (95% CI 6.79–7.01). When assessing fibrosis using Criteria 2–5, among the overall study cohort, prevalence of F3 fibrosis ranged from 0.35% - 2.41% and the prevalence of F4 fibrosis ranged from 0.71% - 1.21% (Table 2). As with Criteria 1, similar trends were observed with Criteria 2–5, such that increasing number of metabolic co-morbidities were associated with higher prevalence of F3 and F4

**Table 1. Characteristics of the study cohort.**

| Variables | Proportion (%) | Frequency (N) |
|---|---|---|
| Total | | 20,204,004 |
| Sex | | |
| Female | 57.4% | 11,579,119 |
| Male | 42.6% | 8,590,269 |
| Age (mean, SD, Median, Range) | 54.6 ± 17.3 | 55.7 (18–99) |
| Elevated ALT | 7.4% | 1,492,650 |
| Elevated Triglycerides | 29.0% | 4,788,095 |
| Elevated Hemoglobin A1C | 19.9% | 1,966,854 |
| Decreased HDL | 24.3% | 3,999,546 |
| Hypertension | 39.1% | 7,907,090 |

Note:

Elevated ALT: > 25 U/L in women, > 35 U/L in men

Elevated Triglycerides: > = 150 mg/dL

Elevated Hemoglobin A1C: > = 6.5%

Decreased HDL: <40mg/dL in men, <50 mg/dL in women

Hypertension: ICD 9/10 code for hypertension

**Table 2. Prevalence of F3 and F4 fibrosis by different FIB-4 criteria.**

| | | Overall Cohort | | NAFLD/NASH ICD-9/10 Cohort | |
|---|---|---|---|---|---|
| | | N = 20,204,004 | | N = 208,749 | |
| | FIB-4 Categories | Prevalence (%) | 95% CI | Prevalence (%) | 95% CI |
| Criteria 1 | F3/F4 > 2.67 | 3.12% | 3.11–3.13% | 6.90% | 6.79–7.01% |
| Criteria 2 | F4 > 4.12 | 0.71% | 0.71–0.71% | 2.52% | 2.45–2.59% |
| | 2.67 < F3 < = 4.12 | 2.41% | 2.40–2.42% | 4.38% | 4.29–4.47% |
| Criteria 3 | F4 > 3.5 | 1.21% | 1.20–1.21% | 3.67% | 3.59–3.75% |
| | 2.67 < F3 < = 3.5 | 1.91% | 1.91–1.92% | 3.23% | 3.16–3.31% |
| Criteria 4 | F4 > 3.5 | 1.21% | 1.20–1.21% | 3.67% | 3.59–3.75% |
| | 3.25 < F3 < = 3.5 | 0.35% | 0.35–0.35% | 0.68% | 0.65–0.72% |
| Criteria 5 | F4 > 4.12 | 0.71% | 0.71–0.71% | 2.52% | 2.45–2.59% |
| | 3.25 < F3 < = 4.12 | 0.84% | 0.84–0.85% | 1.83% | 1.77–1.89% |
| | | One Metabolic Co-Morbidity | | Two Metabolic Co-Morbidities | |
| | | N = 231,429 | | N = 208,943 | |
| | FIB-4 Categories | Prevalence (%) | 95% CI | Prevalence (%) | 95% CI |
| Criteria 1 | F3/F4 > 2.67 | 5.07% | 4.98–5.16% | 4.79% | 4.70–4.88% |
| Criteria 2 | F4 > 4.12 | 1.70% | 1.64–1.75% | 1.46% | 1.41–1.51% |
| | 2.67 < F3 < = 4.12 | 3.37% | 3.30–3.44% | 3.33% | 3.25–3.41% |
| Criteria 3 | F4 > 3.5 | 2.56% | 2.50–2.63% | 2.29% | 2.23–2.36% |
| | 2.67 < F3 < = 3.5 | 2.51% | 2.44–2.57% | 2.50% | 2.43–2.57% |
| Criteria 4 | F4 > 3.5 | 2.56% | 2.50–2.63% | 2.29% | 2.23–2.36% |
| | 3.25 < F3 < = 3.5 | 0.51% | 0.48–0.54% | 0.53% | 0.49–0.56% |
| Criteria 5 | F4 > 4.12 | 1.70% | 1.64–1.75% | 1.46% | 1.41–1.51% |
| | 3.25 < F3 < = 4.12 | 1.38% | 1.33–1.43% | 1.35% | 1.30–1.40% |
| | | Three Metabolic Co-Morbidities | | Four Metabolic Co-Morbidities | |
| | | N = 117,797 | | N = 43,035 | |
| | FIB-4 Categories | Prevalence (%) | 95% CI | Prevalence (%) | 95% CI |
| Criteria 1 | F3/F4 > 2.67 | 5.57% | 5.44–5.70% | 6.68% | 6.44–6.92% |
| Criteria 2 | F4 > 4.12 | 1.53% | 1.46–1.60% | 1.62% | 1.50–1.74% |
| | 2.67 < F3 < = 4.12 | 4.04% | 3.93–4.15% | 5.06% | 4.86–5.27% |
| Criteria 3 | F4 > 3.5 | 2.47% | 2.38–2.56% | 2.81% | 2.66–2.97% |
| | 2.67 < F3 < = 3.5 | 3.10% | 3.00–3.20% | 3.87% | 3.69–4.05% |
| Criteria 4 | F4 > 3.5 | 2.47% | 2.38–2.56% | 2.81% | 2.66–2.97% |
| | 3.25 < F3 < = 3.5 | 0.59% | 0.55–0.64% | 0.76% | 0.68–0.84% |
| Criteria 5 | F4 > 4.12 | 1.53% | 1.46–1.60% | 1.62% | 1.50–1.74% |
| | 3.25 < F3 < = 4.12 | 1.53% | 1.46–1.60% | 1.95% | 1.82–2.09% |

fibrosis. Among the NAFLD/NASH ICD-9/10 cohort, prevalence of F3 fibrosis ranged from 0.68% - 4.38% and prevalence of F4 fibrosis ranged from 2.52% - 3.67% using these Criteria (Table 2).

To further assess the impact of concurrent metabolic co-morbidities on prevalence of advanced fibrosis among individuals with NASH, we specifically focused on the subset of NAFLD/NASH ICD-9/10 cohort and calculated the prevalence of F3/F4 fibrosis in patients with elevated ALT and incremental increasing number of concurrent metabolic co-morbidities (Fig 2). When using Criteria 1, the prevalence of F3/F4 fibrosis among NASH patients increased significantly from 7.82% in patients with elevated ALT and one metabolic co-morbidity to 11.63% to NASH patients with elevated ALT and four concurrent metabolic co-

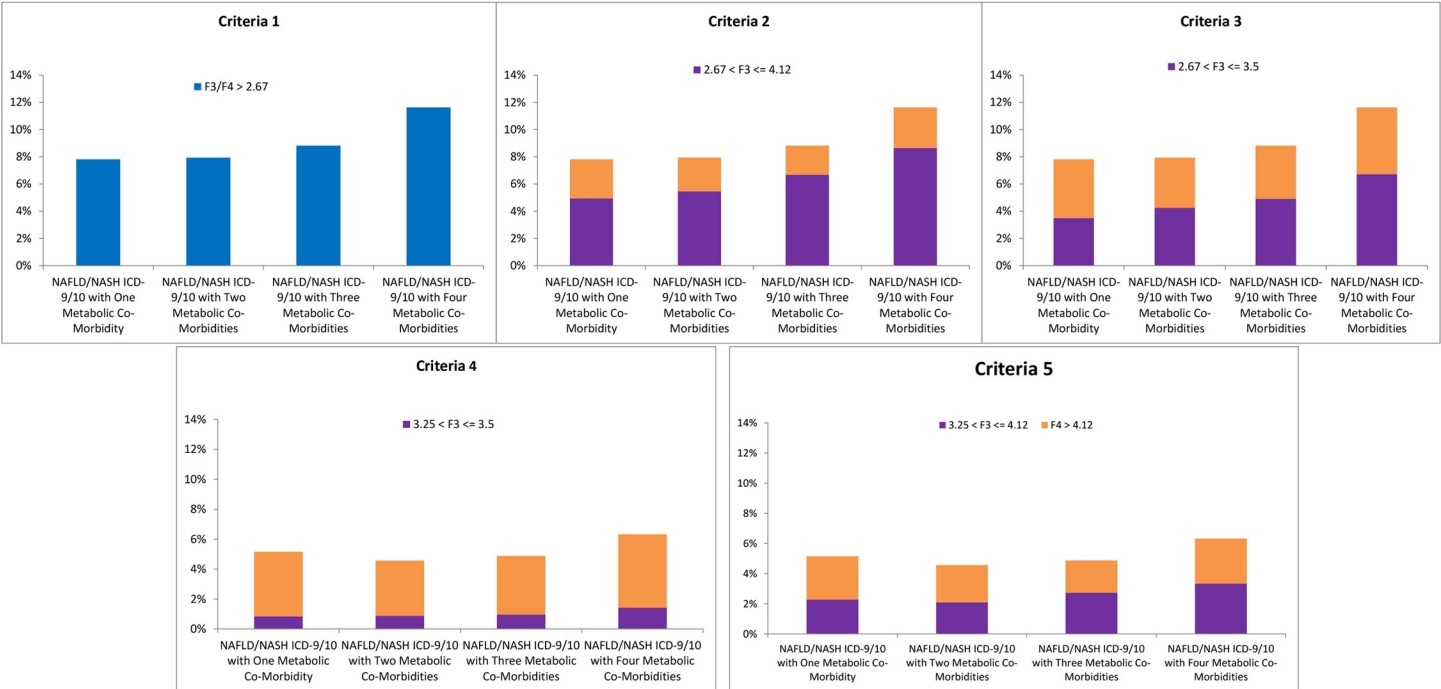

**Fig 2. Prevalence of F3 and F4 fibrosis among NAFLD/NASH ICD-9/10 patients with increasing number of metabolic co-morbidities.**

morbidities, p<0.001. Similar trends in prevalence of F3 and F4 fibrosis were observed when using Criteria 2–5, such that individuals with NASH concurrent with four metabolic co-morbidities had higher prevalence of F3 fibrosis (range, 1.42% - 8.65%) and F4 fibrosis (range, 2.98% - 4.90%) than those with one, two, or three concurrent metabolic co-morbidities (Fig 2).

On adjusted multivariable logistic regression analysis evaluating the odds of F3/F4 fibrosis (Criteria 1) among NAFLD/NASH ICD-9/10 cohort patients, males with NASH had significantly greater odds of F3/F4 fibrosis compared to females with NASH (OR 1.22, 95% CI 1.16–1.27, p<0.001) (Table 3). Increasing age was also associated with higher odds of F3/F4 fibrosis (OR for each 10 years, 2.12, 95% CI 2.08–2.17). Compared to NASH patients without any of our assessed metabolic co-morbidities, increasing number of metabolic comorbidities was associated with increasing odds of F3/F4 fibrosis (Table 3).

**Table 3. Odds of F3 or greater fibrosis using criteria 1 among NAFLD/NASH ICD-9/10 diagnosis code patients.**

|  | Adjusted Odds Ratio | 95% CI | P-Value |
|---|---|---|---|
| **Sex** |  |  |  |
| Female | 1.00 | Reference | - |
| Male | 1.22 | 1.16–1.27 | < 0.001 |
| **Age (each 10 years)** | 2.12 | 2.08–2.17 | < 0.001 |
| **Metabolic Co-Morbidities** |  |  |  |
| None | 1.00 | Reference | - |
| One | 1.18 | 1.08–1.29 | < 0.001 |
| Two | 1.28 | 1.18–1.40 | < 0.001 |
| Three | 1.34 | 1.23–1.47 | < 0.001 |
| Four | 1.56 | 1.40–1.73 | < 0.001 |

## Discussion

Among a large U.S. national clinical laboratory database, when focusing specifically on patients with NAFLD/NASH ICD-9/10 diagnosis codes, prevalence of advanced fibrosis ranged from 4.35% - 6.90%, and the prevalence of F4 fibrosis/cirrhosis ranged from 2.52%–3.67%. Using an estimate of 75 million adults with nonalcoholic fatty liver disease in the U.S., the prevalence of NASH with advanced fibrosis is as high as 5.18 million adults, and the prevalence of NASH cirrhosis is as high as 2.75 million adults.

Given the likely under-diagnosis and thus under-estimate of NASH when relying only on ICD-9/10 coding, we evaluated a larger cohort of patients with available labs to assess ALT and concurrent metabolic co-morbidities after excluding those with HBV or HCV and excluding those with pregnancy or potential alcoholic liver disease, without relying on ICD-9/10 coding. Among this cohort, we observed significantly higher prevalence and risk of advanced fibrosis associated with increasing number of co-morbidities. For example, prevalence of advanced fibrosis in patients with elevated ALT and four concurrent metabolic co-morbidities is as high as 6.68%, and the prevalence of cirrhosis in patients with elevated ALT and four concurrent metabolic co-morbidities is as high as 2.81%. Given the methods employed, it would be a reasonable estimate that these patients with elevated ALT and four concurrent metabolic co-morbidities have NASH, thus making the estimates of advanced fibrosis and cirrhosis prevalence among this group quite concerning.

To our knowledge, this study is one of the first to utilize a large national clinical laboratory database to refine prevalence estimates of patients with NASH and advanced fibrosis. Previous studies evaluating prevalence of NASH and advanced fibrosis have utilized cross sectional survey-based study designs that have may have significantly underestimated true prevalence. For example, three recent studies utilizing data from the National Health and Nutrition Examination Survey (NHANES) evaluated prevalence and predictors of advanced fibrosis among patients with suspected NAFLD based on algorithms that combined patient reported history and co-morbidities and serological biomarkers[9, 10, 19]. Using the U.S. fatty liver index to identify adults with NAFLD and NAFLD fibrosis score to assess for presence of fibrosis, Le, et al. reported a 30.0% prevalence of NAFLD among U.S. adults from 1999–2012 NHANES, among which 10.3% had advanced fibrosis[19]. Similarly, Kabbany, which focused on the latter 2009–2012 NHANES dataset, observed a prevalence of 0.84–1.75% for NASH advanced fibrosis and prevalence of 0.178% for NASH cirrhosis[10]. Most recently, Wong, et al focused on the 2011–2014 NHANES reported 21.9% prevalence of NAFLD among U.S. adults, among which 2.3–9.7% had advanced fibrosis, corresponding to 5.0 million adults[9]. While these previous studies provide some estimates of NASH with advanced fibrosis, they are likely underestimates given the limitations of NHANES, including inherent biases from survey-based methods such as recall bias or reporter bias. Furthermore, misclassification bias due to reliance of surrogates of disease states to identify NASH may have affected prevalence estimates. Our current study utilizes national clinical laboratory data that captures a large proportion of the U.S. adult population and utilized objective serology based measures for identifying NASH with advanced fibrosis, which improved the generalizability and accuracy of our results.

Given that development of hepatic fibrosis and advanced fibrosis in particular is the strongest predictor of liver-related outcomes and mortality in patients with NASH[6, 8, 20], understanding which NASH populations are at greatest risk of progression to advanced fibrosis is of critical clinical significance. Focusing on four components of the metabolic syndrome, our study demonstrated increasing number of concurrent metabolic co-morbidities to be associated with increasing risks for advanced fibrosis in NASH patients. The detrimental impact of

metabolic disease is well known, and Stepanova, et al. using the NHANES data observed that concurrent metabolic syndrome was associated with increased liver-related and all-cause mortality in patients with chronic liver disease[21]. Similarly, Younossi, et al. observed that metabolic syndrome among NAFLD patients in particular, was also associated with increased risk of overall, liver-related, and cardiovascular mortality[22]. Insulin resistance and diabetes mellitus have demonstrated strong associations with increased risks of advanced fibrosis in NASH patients[23–27]. For example, Petta, et al. evaluated 863 patients with biopsy proven NAFLD and observed that among patients < 55 years, the risk of advanced fibrosis was primarily driven by insulin resistance and visceral adiposity, whereas risk of advanced fibrosis in NAFLD patients 55 years and older, low HDL and insulin resistance were the main factors [24]. In our current study, we focused on three laboratory biomarker-biomarker based metabolic abnormalities–HDL, TG, and hemoglobin A1C, given that these are specific components of the metabolic syndrome. Data for waist circumference and presence of hypertension (the other two variables in the metabolic syndrome definition) were not available for assessment. Among our NASH cohort, we observed significantly higher risk of advanced fibrosis associated with presence of metabolic co-morbidities, with the highest risk group (those with four concurrent metabolic abnormalities) having a 56% higher risk of advanced fibrosis compared to patients without any metabolic co-morbidities. These are clinically significant observations and highlight the importance of focusing on optimizing the management of metabolic co-morbidities among NASH patients to prevent further disease progression.

As this is one of the first studies to utilize national clinical laboratory data to provide estimates of NASH with advanced fibrosis prevalence among U.S. adults, our observations provide important epidemiology data to guide healthcare resource planning. The dataset used captures a large sample of U.S. adults undergoing routine laboratory testing, which improves the generalizability of our study. While our overall study cohort identified probable NASH patients primarily through exclusion of chronic HBV, chronic HCV and exclusion of patients who were pregnant or had potential alcoholic liver disease (based on ICD-9/10 coding), we acknowledge the limitation of not having specific data on alcohol consumption. To address this limitation, we performed parallel analyses using similar exclusion criteria that specifically focused on patients with NAFLD/NASH ICD-9/10 diagnosis codes and performed similar assessments of F3 and F4 prevalence. While we utilized objective laboratory criteria to identify metabolic abnormalities, it is possible that some patients were on lipid-lowering therapies or diabetes medications that may have contributed to normal laboratory values, which would have contributed to an underestimation of the prevalence of metabolic abnormalities in this subset. However, this limitation likely biased our estimates towards a more conservative estimate of prevalence which is preferred over the potential type 1 error of overestimation. Despite these limitations, our current data provide important generalizable data on prevalence and predictors of NASH advanced fibrosis among U.S. adults.

In conclusion, among a large nationally representative clinical laboratory database of U.S. adults, the prevalence of NASH with advanced fibrosis was as high as 6.68% and the prevalence of NASH with cirrhosis was as high as 2.81%, which represents 5.18 million and 2.75 million adults, respectively when using an estimate of 75 million U.S. adults with NAFLD.[1] Co-morbid metabolic abnormalities were associated with significantly higher risk of advanced fibrosis among NASH patients. These data highlight the significant clinical and economic impact that NASH will play on healthcare systems.

## Author Contributions

**Conceptualization:** Robert J. Wong, Tram Tran, Harvey Kaufman, Robert Gish.

**Data curation:** Robert J. Wong, Harvey Kaufman, Justin Niles.

**Formal analysis:** Robert J. Wong, Tram Tran, Harvey Kaufman, Justin Niles, Robert Gish.

**Funding acquisition:** Robert J. Wong, Robert Gish.

**Investigation:** Robert J. Wong, Tram Tran, Robert Gish.

**Methodology:** Robert J. Wong, Justin Niles, Robert Gish.

**Project administration:** Robert J. Wong.

**Supervision:** Robert Gish.

**Writing – original draft:** Robert J. Wong.

**Writing – review & editing:** Robert J. Wong, Tram Tran, Harvey Kaufman, Justin Niles, Robert Gish.

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
