## [Decision Letter · Decision Letter 0]

18 Jun 2019

PONE-D-19-14625

Increasing Metabolic Co-Morbidities are Associated with Higher Risk of Advanced Fibrosis in Nonalcoholic Steatohepatitis

PLOS ONE

Dear Dr Wong,

Thank you for submitting your manuscript to PLOS ONE. After careful consideration, we feel that it has merit but does not fully meet PLOS ONE’s publication criteria as it currently stands. Therefore, we invite you to submit a revised version of the manuscript that addresses the points raised during the review process.

We would appreciate receiving your revised manuscript by Aug 02 2019 11:59PM. To enhance the reproducibility of your results, we recommend that if applicable you deposit your laboratory protocols in protocols.io, where a protocol can be assigned its own identifier (DOI) such that it can be cited independently in the future. For instructions see: http://journals.plos.org/plosone/s/submission-guidelines#loc-laboratory-protocols

We look forward to receiving your revised manuscript.

Kind regards,

Ming-Lung Yu, MD, PhD

Academic Editor

PLOS ONE

Journal Requirements:

2. Please ensure that the algorithm used is described in enough detail to meet PLOS ONE criteria for reproducibility (https://journals.plos.org/plosone/s/criteria-for-publication#loc-3)

I have read the journal's policy and the authors of this manuscript have the following competing interests:

Robert Wong: research grants, consulting, advisory board, speaker’s bureau – Gilead Sciences; research grants – Abbvie; speaker’s bureau – Salix, Bayer; research grant – AASLD Foundation

Tram Tran: employee and stocks – Gilead Sciences

Harvey Kaufman: None

Justin Niles: None

Robert Gish: consulting, advisor - Abbot, AbbVie, Alexion, Arrowhead, Bayer AG, Biocollections, Bristol-Myers Squibb Company, Contravir, Eiger, Enyo, eStudySite, Genentech, Gilead Sciences, HepaTX, HepQuant, Hoffmann-LaRoche Ltd., Intellia, Intercept, Ionis Pharmaceuticals, Janssen, MedImmune, Merck, Prometheus, Quest, Shionogi, Transgene, Trimaran. Scientific/clinical Advisory Boards -AbbVie, Merck, Arrowhead, Bayer, Contravir, Dova Pharmaceuticals, Eiger, Enyo, Janssen, Medimmune, Janssen/J&J, Intercept, Shionogi, Spring Bank. Clinical trials – eStudySite. Chair Clinical Advisory Board – Arrowhead. Data Safety Monitoring Board – Ionis. Speaker’s bureau – AbbVie, Alexion, Bayer, BMS, Gilead Sciences Inc., Merck. Minor stock shareholder - Athenex, Triact, Synageva, RiboSciences, CoCrystal. Stock options – Arrowhead, Eiger.

We note that you received funding from commercial sources.

Reviewers' comments:

Reviewer's Responses to Questions

**Comments to the Author**

1. Is the manuscript technically sound, and do the data support the conclusions?

Reviewer #1: Yes

Reviewer #2: Partly

2. Has the statistical analysis been performed appropriately and rigorously? 

Reviewer #1: Yes

Reviewer #2: Yes

3. Have the authors made all data underlying the findings in their manuscript fully available?

Reviewer #1: Yes

Reviewer #2: Yes

4. Is the manuscript presented in an intelligible fashion and written in standard English?

Reviewer #1: Yes

Reviewer #2: Yes

5. Review Comments to the Author

Reviewer #1: This study was performed by a group of experts, aiming to utilize a nationally representative U.S. clinical laboratory database to evaluate prevalence of advanced fibrosis using a systematic algorithm of non-invasive fibrosis biomarker (FIB-4) and to evaluate the impact of metabolic co-morbidities on odds of advanced fibrosis. The findings are important. It would be even nice to improve the quality of figures by using colour figures of higher resolution.

Reviewer #2: The Authors conducted the retrospective study aiming to evaluate prevalence of advanced fibrosis and to evaluate the impact of metabolic co-morbidities on odds of advanced fibrosis based on a large-scale database. They found that the prevalence of advanced fibrosis ranged from 6.18% - 9.07%, and the prevalence of F4 fibrosis/cirrhosis ranged from 3.91% – 5.35%. They concluded that co-morbid metabolic abnormalities were associated with higher risk of advanced fibrosis among NASH patients. Although the study touched the important issue indicating the parallel association between disease severity and metabolic abnormalities in NAFLD patients, some points need clarification.

1. The definition of NAFLD/NASH in the current study was based on the ICD 9/10 diagnosis coding. The validation of the diagnosis should at least be performed to its best. The exclusion of alcoholism, pregnancy-related fatty change could be the confounding factors.

2. The Authors used FIB-4 as the non-invasive serum panel for fibrosis assessment. The performance could be validated by APRI.

3. Hypertension is one of the main components of metabolic abnormalities. How come the results if the diagnosis coding of hypertension were added to enhance the accuracy of metabolic syndrome in the current study.

4. The conclusion depicted "prevalence of NASH ..., representing 6.80 million and 4.01 million, respectively." , which could not be sufficiently supported by the results. A mild form is suggested.

6. PLOS authors have the option to publish the peer review history of their article (what does this mean?). If published, this will include your full peer review and any attached files.

Reviewer #1: No

Reviewer #2: No

---

## [Author Response · Author response to Decision Letter 0]

2 Jul 2019

To:

Dr. Joerg Heber

Editor-in-Chief, PLOS ONE

Dr. Ming-Lung Yu

RE: Manuscript (PONE-D-19-14625)

" Increasing Metabolic Co-Morbidities are Associated with Higher Risk of Advanced Fibrosis in Nonalcoholic Steatohepatitis"

Dear Dr. Yu and Dr. Heber

My co-authors and I are grateful of the reviewers’ comments and your continued consideration of our manuscript for publication in PLOS ONE. Please find below our point-by-point responses to the reviewers’ comments and suggestions. We have also made revisions in the manuscript text to incorporate the changes resulting from these reviewer comments. We have included two versions of our revised manuscript – track changes and clean version - as per your instructions.

Comments to the Author(s):

Reviewer #1: This study was performed by a group of experts, aiming to utilize a nationally representative U.S. clinical laboratory database to evaluate prevalence of advanced fibrosis using a systematic algorithm of non-invasive fibrosis biomarker (FIB-4) and to evaluate the impact of metabolic co-morbidities on odds of advanced fibrosis. The findings are important. It would be even nice to improve the quality of figures by using colour figures of higher resolution.

Response:

Thank you for this suggestion – we have revised our figures to be of color and improved the resolution of our submitted figure files.

Reviewer #2: The Authors conducted the retrospective study aiming to evaluate prevalence of advanced fibrosis and to evaluate the impact of metabolic co-morbidities on odds of advanced fibrosis based on a large-scale database. They found that the prevalence of advanced fibrosis ranged from 6.18% - 9.07%, and the prevalence of F4 fibrosis/cirrhosis ranged from 3.91% – 5.35%. They concluded that co-morbid metabolic abnormalities were associated with higher risk of advanced fibrosis among NASH patients. Although the study touched the important issue indicating the parallel association between disease severity and metabolic abnormalities in NAFLD patients, some points need clarification.

1. The definition of NAFLD/NASH in the current study was based on the ICD 9/10 diagnosis coding. The validation of the diagnosis should at least be performed to its best. The exclusion of alcoholism, pregnancy-related fatty change could be the confounding factors.

Response:

Thank you for this very important question. This is one of the stated limitations of this database in that our diagnosis and identification of NAFLD/NASH relies on the data that we have available in the current dataset. We tried to utilize a systematic algorithm, and agree with this reviewer that even among the cohort of patients with ICD-9/10 codes, we were not able to perfectly exclude all other potential concurrent disease states. For example, we did not have data on alcohol consumption and thus were not able to accurately capture this. However, based on this reviewer’s very insightful feedback, we did go back and incorporate two additional levels of exclusion criteria to further validate our study findings. We implemented exclusion criteria of any patients with ICD-9/10 codes for alcohol use/abuse or alcoholic liver disease. We further reviewed the Quest Diagnostics database and were able to exclude any individual with evidence if pregnancy. We have completely revised our entire study, which is reflected in an updated figure 1 flow chart as well as updated all of the results and relevant methods section. Thank you again for this feedback as we feel that the revisions arising out of this suggestion has truly strengthened our manuscript.

Reviewer #2:

2. The Authors used FIB-4 as the non-invasive serum panel for fibrosis assessment. The performance could be validated by APRI.

Response:

Thank you for this suggestion. While we did in fact perform similar analyses using APRI, we chose not to include this in our analyses. Firstly, when evaluating the performance characteristics of different non-invasive methods for assessing hepatic fibrosis, FIB-4 has significantly better performance characteristics (higher AUROC) than APRI, and thus our analyses focused on FIB-4 given the better sensitivity and specificity (Xiao, et al. Hepatology 2017;66:1486-1501). Furthermore, recent studies demonstrated that the performance characteristics of these non-invasive markers are affected by age, and a recent study by McPherson et al. (Am J Gastroenterol 2017;112:740-51) developed a revised cut-off of FIB-4>2.67, which we were able to incorporate into our analyses. However, no such analyses have been evaluated with APRI, and thus the performance characteristics may be worse than originally thought. Secondly, our analysis was specifically focused on evaluating patients with F3 or greater fibrosis, which can be done with the FIB-4 based algorithm that we illustrated. However, currently validated cutpoints for APRI are only validated for assessing at the F2 and F4 levels, and have not specifically been validated for the F3 fibrosis level. For these reasons and limitations, we felt strongly about not including the APRI data in our manuscript. However, the data that we did analyze demonstrated similar trends, such that increasing number of metabolic co-morbidities also correlated with increasing risk of hepatic fibrosis when using APRI-based cutpoints.

Reviewer #2: 

3. Hypertension is one of the main components of metabolic abnormalities. How come the results if the diagnosis coding of hypertension were added to enhance the accuracy of metabolic syndrome in the current study.

Response:

Thank you for this comment. This is a great suggestion. We have completely revised our entire analysis and updated our manuscript to include hypertension as a fourth metabolic co-morbidity analyzed. Given that blood pressure measurements and medication data were not available for analysis, we utilized ICD=9/10 coding to identify patient with hypertension. Thank you for this important suggestion as you can see by adding hypertension, it actually strengthened our analysis of the correlation between metabolic co-morbidities and the presence of advanced fibrosis in the multivariate model.

Reviewer #2:

4. The conclusion depicted "prevalence of NASH ..., representing 6.80 million and 4.01 million, respectively." , which could not be sufficiently supported by the results. A mild form is suggested.

Response:

Thank you for raising this important point. These estimates were based on a estimated national NAFLD prevalence of 75 million based on a recent systematic review (Rinella ME. JAMA. 2015;313(22):2263-73). We neglected to mention this 75 million estimate as our basis. We have revised our conclusion section incorporate this.

We thank the editors and reviewers for the helpful comments and suggestions that have improved our manuscript significantly and have made our manuscript more valuable to the readers of PLOS ONE. We hope that you will find the response to the above comments and the corresponding manuscript revisions acceptable. Thank you again for the opportunity to revise our manuscript for your continued consideration.

Sincerely,

Robert J. Wong, M.D., M.S., F.A.C.G.

Assistant Clinical Professor of Medicine

Division of Gastroenterology and Hepatology

Alameda Health System – Highland Hospital

Highland Care Pavilion – 5th Floor, Endoscopy Unit

1411 East 31st Street

Oakland, CA 94602

Phone: 510-437-6531

Email: rowong@alamedahealthsystem.org

---

## [Decision Letter · Decision Letter 1]

22 Jul 2019

Increasing Metabolic Co-Morbidities are Associated with Higher Risk of Advanced Fibrosis in Nonalcoholic Steatohepatitis

PONE-D-19-14625R1

Dear Dr. Wong,

We are pleased to inform you that your manuscript has been judged scientifically suitable for publication and will be formally accepted for publication once it complies with all outstanding technical requirements.

With kind regards,

Ming-Lung Yu, MD, PhD

Academic Editor

PLOS ONE

Additional Editor Comments (optional):

Reviewers' comments:

Reviewer's Responses to Questions

**Comments to the Author**

1. If the authors have adequately addressed your comments raised in a previous round of review and you feel that this manuscript is now acceptable for publication, you may indicate that here to bypass the “Comments to the Author” section, enter your conflict of interest statement in the “Confidential to Editor” section, and submit your "Accept" recommendation.

Reviewer #1: All comments have been addressed

Reviewer #2: All comments have been addressed

2. Is the manuscript technically sound, and do the data support the conclusions?

Reviewer #1: Yes

Reviewer #2: (No Response)

3. Has the statistical analysis been performed appropriately and rigorously? 

Reviewer #1: Yes

Reviewer #2: (No Response)

4. Have the authors made all data underlying the findings in their manuscript fully available?

Reviewer #1: Yes

Reviewer #2: (No Response)

5. Is the manuscript presented in an intelligible fashion and written in standard English?

Reviewer #1: Yes

Reviewer #2: (No Response)

6. Review Comments to the Author

Reviewer #1: I have no additional comments.

Reviewer #2: (No Response)

7. PLOS authors have the option to publish the peer review history of their article (what does this mean?). If published, this will include your full peer review and any attached files.

Reviewer #1: No

Reviewer #2: Yes: Jee-Fu Huang

---

## [Editor Report · Acceptance letter]

24 Jul 2019

PONE-D-19-14625R1 

Increasing Metabolic Co-Morbidities are Associated with Higher Risk of Advanced Fibrosis in Nonalcoholic Steatohepatitis 

Dear Dr. Wong:

I am pleased to inform you that your manuscript has been deemed suitable for publication in PLOS ONE. Congratulations! Your manuscript is now with our production department. 

With kind regards,

on behalf of

Dr. Ming-Lung Yu 

Academic Editor

PLOS ONE